# Antioxidant Activity of Graphene Quantum Dots Prepared in Different Electrolyte Environments

**DOI:** 10.3390/nano9121708

**Published:** 2019-11-29

**Authors:** Lin Zhao, Yingmin Wang, Yan Li

**Affiliations:** School of Materials Science and Engineering, University of Science and Technology Beijing, Beijing 100083, China; linluluxiu@163.com (L.Z.); 18811395537@163.com (Y.W.)

**Keywords:** GQDs, free radical scavenging, antioxidant, electrochemical method

## Abstract

Antioxidants can reduce or inhibit damage such as oxidative decay caused by elevated levels of free radicals. Therefore, pursuing antioxidants with excellent properties has attracted more and more attention. Graphene quantum dots (GQDs) are considered a promising material because of their good free radical scavenging activity, low toxicity, and excellent water solubility. However, their scavenging efficiency, antioxidant mechanism, and effective control methods need to be improved. Herein, in order to further reveal the antioxidant mechanism of GQDs, the role of electrolytes in improving the antioxidant activity of GQDs is explored. In addition, 1,1-diphenyl-2-picrazine (DPPH∙), hydroxyl (∙OH), and superoxide (∙O_2_^−^) free radicals are used to evaluate the antioxidant activity of the as-prepared GQDs. Combined with transmission electron microscopy, Fourier-transform infrared spectroscopy, Raman spectroscopy, and cyclic volt–ampere characteristic curves, the effects of an electrolytic environment on the surface functional groups, charge transfer capability, and defect states of GQDs are obtained. The antioxidant mechanism of GQDs and how to improve their antioxidant activity are further elucidated.

## 1. Introduction

Reactive oxygen species (ROS), including free radicals such as hydroxyl (∙OH), superoxide (∙O_2_^−^), alkoxy (RO∙), and peroxide (ROO∙), are byproducts of cellular redox processes. ROS can contain one or more unpaired electron and are, therefore, highly reactive [1,2,3,4]. In general, ROS play a dual role in biological systems. If they remain at a proper level, they are involved in a variety of physiological effects and many cellular signaling processes. When they are in an excessive amount, they can have deleterious effects on biological systems [5,6], such as destroying the DNA, proteins, and lipids of living organisms, leading to various inflammations and diseases [7]. Antioxidants can eliminate ROS or reduce them to a proper level, thus maintaining normal function of biological systems and alleviating the development of disease. Developing rational and efficient antioxidant materials is a critical issue. Interaction between antioxidants and ROS is mainly by hydrogen atom transfer (HAT), single electron transfer (SET), and transition metal chelation [8,9]. 

As a quasi-zero-dimensional carbon nanomaterial, graphene quantum dots (GQDs) have stable fluorescence [10,11,12], water solubility [13], excellent biocompatibility [14,15], and low cytotoxicity [16,17,18]. Thus, it exhibits great application potential in the fields of bio-imaging [19], catalysis [20,21], desalination [22,23,24], plasmonics [25,26,27,28], and sensing [29]. Moreover, the antioxidant activity of GQDs has received much attention [1,30]. Although recent studies have suggested that some surface-modified GQDs may have genotoxicity to organisms [31,32,33,34,35], they may limit the application potential of GQDs as an antioxidant to some extent. However, this genotoxicity is associated with the dose, surface modification, and oxidative stress by light-induced and related factors. Studies on the antioxidant activity of GQDs and related antioxidant mechanisms are still of great significance and value for the treatment of diseases, especially those caused directly or indirectly by free radicals [4]. Therefore, many efforts have reported the antioxidant activity of GQDs and its related mechanisms. Hydrogen donor behavior, sp^2^ hybrid carbon domains, unpaired electrons caused by defects and vacancies, electron transport, types of surface functional groups, and types of doping elements are factors affecting the antioxidant activity of GQDs [1,36,37,38,39,40,41]. However, there are still some issues that need to be addressed, such as the clarity of the antioxidant mechanism and optimization and flexible control of the antioxidant activity of GQDs [42,43,44].

The electrochemical preparation of GQDs has several advantages, such as simplicity, no harsh reaction conditions, and suitability for large-scale synthesis [45,46,47,48]. Different types of GQDs (such as doped, functionalized GQDs) were prepared by electrochemical methods in our group [30,42,44]. It was found that electrochemical parameters have an important influence on the formation of GQDs and can effectively control their physical and chemical properties. Especially, the electrolytic environment has an important impact on composition, doping, and the type and content of the surface functional groups of GQDs. These aspects play a decisive role in the magnitude of the antioxidant activity of GQDs. However, there is no relevant research on the effect and mechanism of electrolyte environments on the antioxidant activity of GQDs. Therefore, in this paper, electrolyte environment effects on the size, surface groups, electron transport ability, and defects of GQDs were investigated. In addition, the antioxidant activity of GQDs prepared in different electrolyte environments was assessed by 1,1-diphenyl-2-picrazine (DPPH∙), ∙OH, and ∙O_2_^−^ free radicals. GQDs with a high antioxidant activity were further analyzed through Fourier-transform infrared spectroscopy (FT-IR), cyclic voltammetry (CV) curves, and Raman spectroscopy. These efforts help to further understand the antioxidant mechanism of GQDs and improve the antioxidant activity of GQDs.

## 2. Materials and Methods 

### 2.1. Synthesis of GQDs 

GQDs were prepared by the constant potential method using a CHI 660D electrochemical workstation. A high-purity graphite rod and a platinum wire were selected as the working and counter electrode, respectively, and the electrolyte was 0.1 M phosphate buffer saline (PBS). The scan voltage, scan rate, and time were 5 V, 0.1 V/s and 48 h, respectively. After scanning, the solution was filtered through a 0.22 µm aqueous filter to remove large-sized carbon particles and dialyzed for 6 days in a 3500 Da molecular-weight-cutoff dialysis bag to remove the attached electrolyte. Finally, a relatively pure GQD aqueous solution was obtained. The final concentration was 0.12 mg/mL by weighing a volume of GQDs before and after drying, and designated GQDs-PBS. The above 0.1 M PBS solution was changed to a 0.1 M NaOH solution and 0.1 M KCl solution, and the other experimental conditions were unchanged. The prepared GQDs were named GQDs-NaOH and GQDs-KCl, respectively. The concentrations of the three GQD solutions were adjusted to the same for subsequent experiments.

### 2.2. Free Radical Scavenging Assay

**(1) DPPH**∙ **scavenging assay:** DPPH∙ is a stable organic free radical that can be used to measure the antioxidant activity of GQDs. At the same time, the DPPH method has the advantages of simple operation and high sensitivity. DPPH∙ can bind to the H atom in the antioxidant to form a stable H-DPPH complex with a change in color, and the concentration of DPPH∙ can be monitored by the characteristic absorption peak at 515 nm [49]. We take 1 mL of 0.1 mg/mL of three different GQDs and mix them well with 1 mL of 0.05 mg/mL DPPH ethanol solution. After standing for different periods of time in the dark, the absorption peak at 515 nm was measured by ultraviolet absorption spectroscopy to reflect the concentration of the remaining DPPH∙ radical. In this experiment, the deionized water solution was used as a blank control group, and the scavenging efficiency (C) of the three kinds of GQDs for DPPH∙ radicals was calculated as follows: C=(1-A_i_/A_0_)/100; A_i_ refers to the absorbance of free radical solution containing samples, while A_0_ refers to the absorbance of the free radical solution excluding samples. 

**(2)** ∙**OH scavenging assay:** Under ultraviolet light, TiO_2_ will produce ∙OH, and the added terephthalic acid can capture ∙OH to form 2-hydroxyterephthalic acid [50]. We demonstrated the activity of antioxidants to scavenge the ∙OH radical by detecting the intensity of fluorescence after the reaction. Experimental procedure: A 2 mL solution contained 25 mM PBS, 0.5 mM terephthalic acid, 50 μg/mL TiO_2_, and 50 μg/mL GQDs. After 1 h of ultraviolet light irradiation (8 W, 365 nm), the emission spectrum of the solution was tested. The final result of the test subtracted the fluorescence produced by the quantum dots themselves. The scavenging efficiency (C) of the three kinds of GQDs for ∙OH radicals was calculated in the same manner as the DPPH∙ scavenging experiment.

**(3)** ∙**O_2_^−^ scavenging assay:** ∙O_2_^−^ is an intermediate in oxygen metabolism, and an excessive amount of ∙O_2_^−^ has a certain toxic effect on the matrix. Lezaic et al. studied the activity of polyaniline tannate (PANI-TA) solid microspheres as antioxidants by cyclic voltammetry [51]. The electrocatalytic reduction of O_2_ to ∙O_2_^−^ was used, and the change in electrochemical response of PANI-TA as a radical scavenger was analyzed. Similarly, in this experiment, ∙O_2_^−^ radicals were generated by dissolving potassium superoxide (KO_2_) in dimethyl sulfoxide (DMSO). Specifically, a 2 mL sample contained 25 mM BMPO, 10% DMSO, 2.5 mM KO_2_, 0.35 mM 18-crown-6, 10 mM PBS buffer (pH = 7.27), and three different GQDs at the same concentration. After 1 min of incubation, the electron spin resonance (ESR) spectrum was measured.

### 2.3. Characterization

The morphology of the samples was characterized by transmission electron microscopy TEM (JEM-2010, JEOL, Tokyo, Japan). The FT-IR spectrum was measured using a NEXUS spectrometer 670 (NEXUS-670, NEXUS, Beijing, China). UV-vis spectra were obtained using a UNICO-2800 spectrophotometer (UV-2800, UNICO, Madison, WI, USA). The GQDs were characterized using a RM 2000 micro-confocal Raman spectrometer (RM-2000, Renishaw, London, England) with 632.8 nm excitation. Cyclic voltammetry (CV) testing was performed on a CHI 660D electrochemical working station (CHI660 D, Chenhua Instrument, Shanghai, China). The ESR spectrum was measured by an electron magnetic resonance measuring instrument (JES-FA200, JEOL, Tokyo, Japan).

## 3. Results and Discussion

TEM images of the three GQDs in Figure 1a clearly illustrate the formation of ultra-small GQDs with good dispersibility and similar spherical shapes, which is similar to the GQDs previouslyreported [52]. Figure 1b further reveals the particle size of GQDs with sizes roughly distributed between 1 and 9 nm. By statistically distributing the particle size of all the QDs in the figure, it can be known that the average particle diameters of GQDs-PBS, GQDs-NaOH, and GQDs-KCl are 3.19 ± 0.90, 3.76 ± 0.70, and 3.36 ± 0.70 nm, respectively. There is no significant difference in shape or size of the samples, indicating that the three electrolytes have similar effects on GQD formation.

The UV absorption spectrum of the three GQDs in Figure 2a exhibits similar characteristic absorption peaks at 230 and 300 nm, the former being due to the π–π* transition of the electrons in the sp^2^ carbon conjugate structure, while the latter shoulder around 300 nm is attributed to the n-π* transition of the sp^3^ conjugate structure [53,54]. Due to the slight difference in concentration of GQDs prepared by the three electrolytes, the absorption curve will fluctuate up and down. Figure 2b–e expound the time-dependent UV absorption spectra of different GQDs after the reaction with DPPH∙. As the reaction time increases, the absorption peaks of different GQDs near 515 nm gradually decrease, which suggests that all three GQDs have a certain antioxidant activity, but the type of electrolyte affects the antioxidant activity. GQDs-NaOH are more efficient at scavenging DPPH∙ in 5 h, reaching 79.55%, while the scavenging efficiencies of GQDs-PBS and GQDs-KCl are 52.80% and 46.89%, respectively. 

In order to study the antioxidant activity of the prepared GQDs in organisms, ∙OH was selected for antioxidant testing. The terephthalic acid in the solution can scavenge ∙OH to form 2-hydroxyterephthalic acid, which emits strong fluorescence at around 430 nm under excitation light of 315 nm. When GQDs are added to a radical solution, the GQDs can scavenge ∙OH to inhibit the formation of 2-hydroxyterephthalic acid. In Figure 3a, the activity of GQDs to scavenge ∙OH is indirectly reflected by comparing the photoluminescence characteristic peak of 2-hydroxyterephthalic acid at 430 nm under 315 nm photoexcitation. The results show that the scavenging efficiency of ∙OH radicals after one hour of reaction was GQDs-NaOH > GQDs-PBS > GQDs-KCl, which were 68.05%, 54.55%, and 30.11%, respectively.

As free radicals have an odd number of electrons and there are unpaired electrons, they can be used as ESR research objects to detect the types of free radicals and perform quantitative analysis of free radicals. Hereafter, the ESR spectroscopy experiments were conducted to measure the concentration of ∙O_2_^−^ and characterize the antioxidant ability of different GQDs. Figure 3c shows the ESR spectrum of deionized water and three GQDs after reaction with ∙O_2_^−^, which was produced by the dissolution of superoxide (KO_2_) in dimethyl sulfoxide (DMSO). The four peaks in the ESR spectrum are characteristic signals of ∙O_2_^−^, and the intensity of these peaks represents the amount of ∙O_2_^−^. It can be seen that the peak of GQDs-NaOH is still the weakest, implying that the content of the complex formed by ∙O_2_^−^ and DMPO is the least. This elucidates that GQDs-NaOH have the highest scavenging efficiency for ∙O_2_^−^, which is consistent with scavenging experiments of DPPH∙ and ∙OH.

From the above results, it is found that the types of electrochemical electrolyte have a conspicuous influence on the antioxidant activity of GQDs. To further clarify the reasons for this difference in antioxidant activity, we conducted further tests to comprehensively analyze the structural and physical properties of the three GQDs. Figure 4a displays the Raman spectrum of GQDs and illustrates two main peaks at about 1345 cm^−1^ (D band) and 1602 cm^−1^ (G band). The D band is related to the sp^3^ hybrid carbon and the G band comes down to the E2g phonon of the sp^2^ carbon atoms [55,56]. The intensity ratio of the D to G band (I_D_/I_G_) can represent the degree of surface defects of GQDs. The peaks of the D and G peaks of the GQDs prepared from the three electrolytes are discrepant, and the value of I_D_/I_G_ also varies. The I_D_/I_G_ value of GQDs-NaOH is 1.03, which is almost equal to the I_D_/I_G_ value of GQDs-KCl (1.02) and greater than that of GQDs-PBS (0.98). This indicates that there are more surface defects on the surface of GQDs-NaOH, and these surface defects can produce more active sites for scavenging free radicals. 

Furthermore, the FT-IR was conducted for the three GQDs to investigate their surface oxygen groups. The concentration and amount of the GQDs were kept the same. The different characteristic peaks in Figure 4c correspond to the vibration of different surface groups. It shows peaks at 1056, 1628, 2399, 2928, and 3421 cm^−1^, which are attributed to C–O–C, –C=O, CO_2_, C–H, and C–OH, respectively. It is also worth noting that the content of surface oxygen functional groups of GQDs prepared by different electrolytes is significantly different. The stretching vibration intensity of –C=O in GQDs-NaOH is significantly higher than those of GQDs-PBS and GQDs-KCl. In addition, the C–OH stretching vibration intensity in GQDs-PBS is lower than those in the other two GQDs. The content of C–O–C in GQDs-KCl is higher than those in GQDs-PBS and GQDs-NaOH, which denotes that more hydroxyl groups and carbonyl functional groups are present at the surface and edges of GQDs-NaOH, while more epoxy groups are present on the interlayer or basal planes of GQDs-KCl. This is in line with the XPS results of our previous research [57]. According to previous reports, different kinds of oxygen-containing groups have different antioxidant activities on GQDs [58]. The hydroxyl and carboxyl groups are typically located at the edges or defects of the graphene sheets and, at the unsaturated bonds, can provide hydrogen, thereby becoming a hydrogen donor for scavenging free radicals. Meanwhile, the epoxy groups are usually located between the graphene sheets or the basal planes [44]. The stable bond structure and difficult-to-contact free radicals make them have low antioxidant activity.

CV measurement was performed by using 0.1 M KCl solution containing 0.5 mM K_3_[Fe(CN)_6_] and 0.5 mM K_4_Fe(CN)_6_ as the electrolyte to estimate the electron transfer ability of three GQDs. Figure 4b shows a typical redox reaction on the surface of the electrode, and GQDs-NaOH manifest the highest peak current value among them. In addition, the peak current value varies with the type of electrolyte used in the preparation, which further signals that the charge transfer capability of GQDs is related to the electrolytic environment. The strength of the electron transfer can be judged according to the strength of the redox peak. It is concluded that in the CV curve of GQDs-NaOH, the redox peak is stronger than that of the other two GQDs, which signifies that the electron transfer ability of GQDs-NaOH is stronger.

From the foregoing, GQDs with different antioxidant activities are prepared by selecting three different electrolytes. Changes in the electrolytic environment have little effect on the size and morphology of the GQDs. The type and content of surface groups, electron transport capacity, and defect state are the decisive factors for the antioxidant properties of the three GQDs. GQDs prepared in the electrolytic environment of NaOH have a higher oxidation degree, more surface hydroxyl groups and carbonyl functional groups, and higher defect states, which are beneficial to their antioxidant activity. Meanwhile, in the KCl electrolysis environment, the surface of GQDs has more epoxy groups, which makes them less active in antioxidant activity. In summary, the electrolytic environment changes the content of different oxygen-containing functional groups, the electron transport ability, and the defect state of GQDs, which, in turn, determines the level of antioxidant activity. By adjusting these factors, GQDs with high antioxidant activity were prepared. The method is simple, green, and can effectively control the antioxidant activity of GQDs, which is of great significance for further understanding the antioxidant mechanism of GQDs and improving their antioxidant activity.

## 4. Conclusions

In this work, three different GQDs were prepared by electrochemically selecting the appropriate electrolyte to produce different electrolytic environments. Three different GQDs all have the ability to scavenge free radicals, and the difference in their antioxidant activity is closely related to their intrinsic structure and surface functional groups. It can be concluded that GQDs prepared in the NaOH electrolysis environment have more active oxygen-containing groups, such as hydroxyl groups and carbonyl groups, which can serve as a hydrogen donor to scavenge free radicals, thereby having a higher antioxidant activity. At the same time, when NaOH is used as an electrolyte, the electron transfer ability and defect state of GQDs are also higher, which also enhances their antioxidant activity to some extent. We believe that a further understanding of the antioxidant mechanisms of GQDs will help to better regulate their antioxidant activity and expand their application in biomedical and other fields.

## Figures and Tables

**Figure 1 nanomaterials-09-01708-f001:**
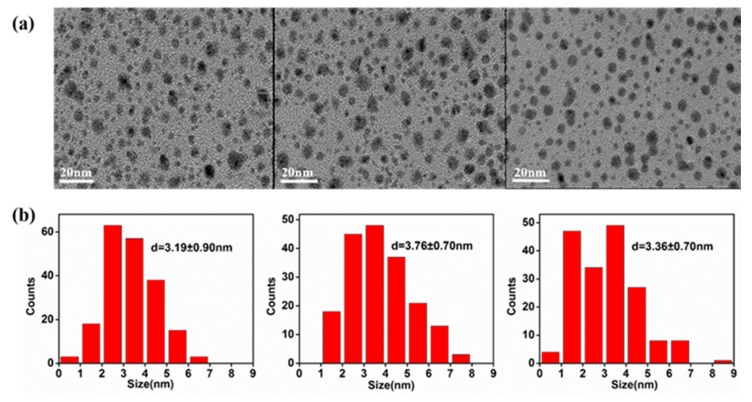
(**a**) TEM image. (**b**) Size distribution image of GQDs-PBS, GQDs-NaOH, and GQDs-KCl.

**Figure 2 nanomaterials-09-01708-f002:**
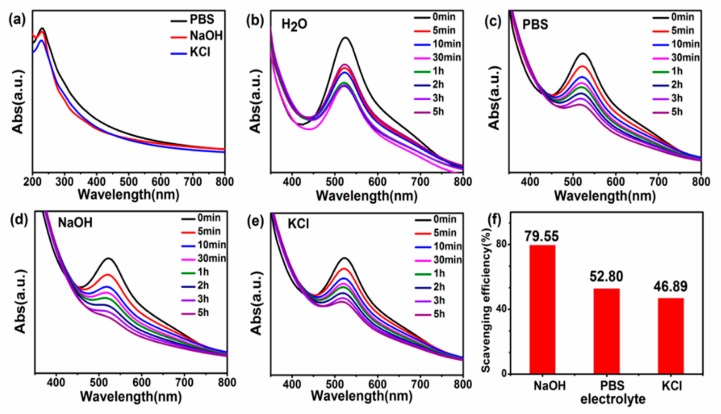
(**a**) Ultraviolet absorption spectrum of GQDs-PBS, GQDs-NaOH, and GQDs-KCl. (**b**–**e**) UV-vis spectra after the reaction. (**f**) Comparison of scavenging efficiency of GQDs prepared with different electrolytes with DPPH∙.

**Figure 3 nanomaterials-09-01708-f003:**
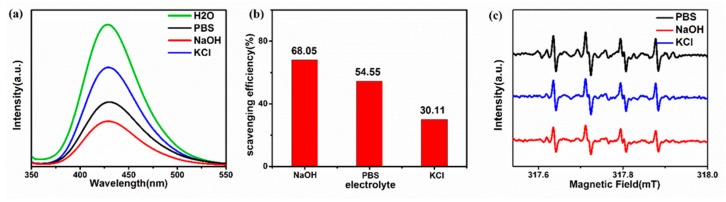
(**a**) Photoluminescence spectrum of GQDs after one hour of reaction with ∙OH radicals. (**b**) ∙OH scavenging efficiency calculated by the amplitude of fluorescence attenuation. (**c**) ESR spectrum of superoxide radicals scavenged by different GQDs after 1 min.

**Figure 4 nanomaterials-09-01708-f004:**
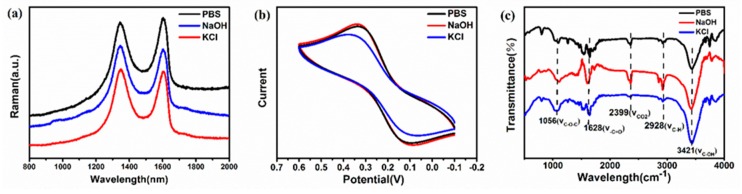
(**a**) Raman spectrum. (**b**) Cyclic voltammetry (CV) characteristics. (**c**) Infrared spectrum of GQDs prepared with different electrolytes.

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
