# Peer review of "Antioxidant Activity of Graphene Quantum Dots Prepared in Different Electrolyte Environments"

_nanomaterials, 2019, doi:10.3390/nano9121708_

Round 1
Reviewer 1 Report
Manuscript is ready for publication
Author Response
Thank you for your comment.
Reviewer 2 Report
The subject matter of this paper deals with antioxidant activity of graphene quantum dots. For this purpose, the authors studied the effects of electrolytic environment on the surface functional groups, charge transfer capability and defect states of graphene quantum dots. They used transmission electron microscopy, Fourier transform infrared spectroscopy, Raman spectroscopy and cyclic volt-ampere characteristic curves. The work is interesting, well organized and comprehensively described. But it needs some corrections for improve.
The authors consider graphene quantum dots are not toxic. However, the potential adverse effects of graphene quantum dots have increasingly attracted attention. Often the toxicity of graphene quantum dots is associated with oxidative stress. This fact limits the potential for their using as antioxidants.
The genotoxicity of graphene quantum dots associated with the generation of reactive oxygen species (ROS) is described in (Xu L, Zhao J, Wang Z. Genotoxic response and damage recovery of macrophages to graphene quantum dots. Sci Total Environ. 2019 May 10;664:536-545. doi: 10.1016/j.scitotenv.2019.01.356.
Hu J, Lin W, Lin B, Wu K, Fan H, Yu Y. Persistent DNA methylation changes in zebrafish following graphene quantum dots exposure in surface chemistry-dependent manner. Ecotoxicol Environ Saf. 2019 Mar;169:370-375. doi: 10.1016/j.ecoenv.2018.11.053.
Xu L, Dai Y, Wang Z, Zhao J, Li F, White JC, Xing B. Graphene quantum dots in alveolar macrophage: uptake-exocytosis, accumulation in nuclei, nuclear responses and DNA cleavage. Part Fibre Toxicol. 2018 Nov 13;15(1):45. doi: 10.1186/s12989-018-0279-8.
Li M, Gu MM, Tian X, Xiao BB, Lu S, Zhu W, Yu L, Shang ZF. Hydroxylated-Graphene Quantum Dots Induce DNA Damage and Disrupt Microtubule Structure in Human Esophageal Epithelial Cells. Toxicol Sci. 2018 Jul 1;164(1):339-352. doi: 10.1093/toxsci/kfy090).
The absorption of graphene quantum dots causes a malfunction of the natural antioxidant system of cells (Deng S, Fu A, Junaid M, Wang Y, Yin Q, Fu C, Liu L, Su DS, Bian WP, Pei DS. Nitrogen-doped graphene quantum dots (N-GQDs) perturb redox-sensitive system via the selective inhibition of antioxidant enzyme activities in zebrafish. Biomaterials. 2019 Jun;206:61-72. doi: 10.1016/j.biomaterials.2019.03.028. Epub 2019 Mar 23.).
The authors should discuss these works.
Author Response
Response: Some studies have shown that GQDs have good biocompatibility and low toxicity, and show great potential in medical applications such as imaging, drug delivery, biosensors and novel therapeutic methods. But by reading these papers, we learned that GQDs have potential genotoxicity. This suggests that GQDs may slightly affect cell viability, which may be related to the dose, duration of action, and oxidative stress. This may limit the application of GQDs as antioxidants to some extent. Based on your suggestion, we have discussed these aspects and cited these important papers in lines 39-44 of the revision. Thank you for your important suggestion.
This manuscript is a resubmission of an earlier submission. The following is a list of the peer review reports and author responses from that submission.
Round 1
Reviewer 1 Report
Line 12-13, use references
Line 14-15 ´´However, the antioxidant mechanism and activity need to be further clarified a improved” For what? Use scientific reasoning.
Line 15 “For electrochemical prepared GQDs,” should read as For electrochemically prepared GQDs”. Please check carefully your grammer usage.
Line 15 . Past tense creates a shift in meaning. Keep using present tense as the rest of the abstract.
Line 16 influence should read as influences. Present tense
Line 17 “Yet, related research rarely reported” Passive voice should read as “ Yet, related research have been rarely reported” Also use references to that claim.
Line 21 “Infrared” why capital?
Line 20-22 poor English usage, sentence is hard to follow.
Line 29-30 Poor English usage, sentence is hard to follow
Line 30 How can ROS cause destructive behavior on variety of diseases? Diseases are already destructive. I think there is again a poor English usage that the authors intend to explain the harmful effects of ROS.
This paper, as a whole, lacks of proper English usage and has serious flaws. Personally I cannot follow the manuscript, I suggest authors to get professional help on scientific writing.
Reviewer 2 Report
The paper by Zot et al. reports on antioxidant activity of graphene quantum dots prepared by various electrolytes. Results reported here are interesting and the manuscript is well written.
The degree of innovation is good, as well as the potential impact. However, some amendments are necessary before publication.
The authors should include among the possible application fields desalination [1-3], plasmonics [4-7] and sensing [8].
The font-size of annotations in Figures should be magnified.
[1] The advent of graphene and other two-dimensional materials in membrane science and technology, Curr. Opin. Chem. Eng. 16 (2017) 78.
[2] Novel graphene quantum dots (GQDs)-incorporated thin film composite (TFC) membranes for forward osmosis (FO) desalination, Desalination 451 (2019) 219.
[3] Graphene quantum dots modified polyvinylidenefluride (PVDF) nanofibrous membranes with enhanced performance for air Gap membrane distillation, Chemical Engineering and Processing-Process Intensification 126 (2018) 222.
[4] Construction of plasmonic Ag and nitrogen-doped graphene quantum dots codecorated ultrathin graphitic carbon nitride nanosheet composites with enhanced photocatalytic activity: full-spectrum response ability and mechanism insight, ACS Appl. Mater. Interfaces 9 (2017) 42816.
[5] Plasmon modes in graphene: status and prospect, Nanoscale 6 (2014) 10927.
[6] Plasmonics in Dirac systems: from graphene to topological insulators, J. Phys.: Condens. Matter 26 (2014) 123201.
[7] Active graphene plasmonics for terahertz device applications, J. Phys. D: Appl. Phys. 47 (2014) 094006.
[8] Development of Graphene Quantum Dots-Based Optical Sensor for Toxic Metal Ion Detection, Sensors 19 (2019) 3850.